# Examining specific emotion dynamics in daily life in male adolescents: An experience sampling method study

Umberto Cauzo[1], Lauriane Constanty[1], Jennifer Glaus[1], Julia Giovannini[1], Marion Abi Kheir[1], Giorgia Miano[1], Caroline Lepage[1], Kerstin Jessica Plessen[1,2], Sébastien Urben[1,2]*

**1** Division of Child and Adolescent Psychiatry, Department of Psychiatry, Lausanne University Hospital (CHUV), Lausanne, Switzerland, **2** Faculty of Biology and Medicine, University of Lausanne, Lausanne, Switzerland

* sebastien.urben@chuv.ch

## Abstract

The dynamics of emotions (e.g., emotion intensity and differentiation) are pivotal for adolescents' mental health. This study examined how psychological characteristics and the social environment influence the dynamics of six emotions in male adolescents during their daily lives. Sixty-two male adolescents (aged between 12–17 years, sufficiently fluent in French, and having access to a smartphone) with varying degrees of adjustment problems (assessed through the youth self-report questionnaire) participated in an experience sampling study (4x/day over 9 days, excluding weekends). Greater adjustment problems, were associated with fewer positive emotions, while greater use of non-adaptive emotion regulation was linked to more negative emotions. Being alone was linked to lower intensity of positive emotions and higher intensity of negative ones. Higher momentary self-control was related to higher intensity of positive emotions and lower intensity of negative ones. This study provides novel insights into the dynamics of male adolescents' emotional experiences by enhancing the understanding of emotions' contextual anchoring combined with psychological characteristics.

## 1. Introduction

Due to the variable nature of our emotional life [1], studying the dynamics and temporal features of emotional experiences provides unique insights into individual differences in mental health [2, 3]. In this study, we aimed to focus on two main features of the dynamics of emotion: emotional intensity (i.e., the strength of an individual's emotional experiences) and emotion differentiation (or emotion granularity), which is the ability to distinguish between various emotional experiences [4, 5]. Emotion dynamics are closely related to mental health and can even serve as a transdiagnostic marker

**Data availability statement:** Some participants did not consent on the re-use of their data, thus we are unable to share the whole dataset. However, we shared the minimal dataset which is available on Zenodo (DOI: 10.5281/zenodo.17341614).

**Funding:** The study was funded by the Swiss National Science Foundation (#CRSK-3_190490 to SU; #32003B_215660 to SU) which we are grateful for. The funders had no role in study design, data collection and analysis, decision to publish, or preparation of the manuscript.

**Competing interests:** The authors have declared that no competing interests exist.

for psychopathologies [6–8], such as adjustment problems [e.g., 9,10]. These adjustment problems include a wide range of difficulties such as internalized symptoms and/or externalized behaviors that affect psychosocial well-being and integration into society [11–14].

Previous studies have shown that, compared to children, adolescents exhibit increased intensity and variability of emotions [15] and use adaptive emotion regulation strategies less frequently than adults [16–19].

From this perspective, emotion regulation and self-control play a significant role in the dynamics of emotions. Emotion regulation specifically refers to processes that modify the dynamics of emotional features, such as the intensity and duration of emotional experiences [20, 21]. These processes can be categorized into non-adaptive (i.e., ineffective, less functional, such as rumination on negative emotions) and adaptive strategies [7]. Moreover, self-control, which is the ability to regulate thoughts and behaviors to improve a specific outcome [22, 23], is closely related to emotion dynamics because it plays an important role in emotion regulation [24].

One of the central aspects of emotions is that they are rooted in a social context, especially as they generally occur in the presence of other people [25, 26]. *Per se*, features of emotion dynamics are influenced by between-person factors (e.g., differences in emotion regulation and self-control skills across individuals), and within-person factors (e.g., various social environments across time within an individual). However, the influence of context (e.g., presence of others, type of social environment) on adolescents' affective dynamics lacks detailed exploration. Previous studies have shown that when alone, youths reported lower arousal or intensity level [27] and more prevalent negative emotions [27, 28]. Additionally, youths in a depressed mood have been shown to spend more time alone and experience more intense negative emotions [28]. Only a few studies [29–31] have observed variations in how individuals differentiate between emotions (i.e., emotion differentiation) over time, and those studies were all conducted among adults. To the best of our knowledge, no studies have been conducted among adolescents, highlighting the need for further studies [32].

To study emotion dynamics at the within-person level, the experience sampling method (ESM; also called ecological momentary assessment or ambulatory assessment) seems a highly appropriate methodology that is anchored in ecological psychology [33]. ESM allows for the regular measurement of an individual's thoughts, experiences, and behaviors in real time in their natural environment, multiple times per day across several days [33, 34]. Compared to studies using retrospective or global self-report measures, this method reflects real-life situations more accurately and minimizes any biases related to recalling past events or psychological states [35, 36]. These advantages lead to increased generalizability and more ecologically valid observations [37].

## 1.1. The current study

Thus, this study aimed to investigate the role of social environment and psychological characteristics in adolescents' emotion dynamics in daily life, at both the

between- and within-person levels. Adopting an ecologically valid approach, we sampled male adolescents' experiences to examine the impact of psychological characteristics (i.e., emotion regulation, self-control, and adjustment problems), the context (e.g., familiar or unfamiliar environments), and the presence of others (alone, with family/friends, or with others) on momentary emotion dynamics features (e.g., intensity and differentiation). Since emotion dynamics differ greatly between males and females [38, 39], the present pilot study focused only on male participants.

## 2. Method

### 2.1. Procedure

The study encompassed a baseline assessment (i.e., trait level, between-person differences) followed by ESM assessments (i.e., state level, within-person variability) administered four times a day (7 a.m., 12 p.m., 4 p.m., 8 p.m.) for nine consecutive days (weekend excluded to ensure a uniform sampling regarding the great variations of daily activities between weekdays and weekend days). Participants received a text message (SMS) with a REDCap® link directly to their personal smartphone to fill out the surveys. For more details about the procedure, see previous publications [see 40,41].

### 2.2. Ethical approval and consent to participate

The study was authorized by the ethics committee of the Canton of Vaud (#2019–02318). Each adolescent first receive oral information about the study, followed by written information for review. Written informed consent is then obtained from each participant before any study-specific procedure. The same process is applied to parents or legal representatives to ensure full understanding and agreement.

### 2.3. Participants

Sixty-two male adolescents aged between 12 and 17 years, with a wide range of adjustment problems, took part in the study (Table 1). All eligible participants had access to a smartphone device and were excluded if their level of French was not sufficient (assessed through verbal intelligence quotient—IQ), hampering the understanding of the questionnaires (to ensure reliable responses). We excluded participants with known diagnoses of schizophrenia, psychosis, or autism spectrum disorders at the time of inclusion in the study and those on antipsychotic medication.

### 2.4. Measures

**2.4.1. Baseline measure—between-person level.** Socioeconomic status (SES) was defined in three categories (i.e., low, medium, and high) at baseline according to the father and mother's highest level of education and work activities [42]. Adjustment problems were assessed by the Youth Self Report (YSR), a self-report measure derived from the Child Behavior Check List (CBCL) [13,14]. To assess adaptive and non-adaptive emotion regulation strategies, we used the Cognitive Emotion Regulation Questionnaire (CERQ) [16]. The Brief Self-Control Scale (BSCS) [43] assessed trait self-control skills at baseline. The Pubertal Development Scale [44] measured the puberty stage at baseline. Finally, we used two subtests (i.e., similarities and vocabulary) of the Wechsler Intelligence Scale for Children [WISC-V; 45] to estimate the verbal IQ at baseline.

**2.4.2. Within-person level. Social environment: context/the presence of others.** The "*context*" variable refers to the momentary environmental context and comprises two categories: "*familiar environment*" or "*external environment.*" It was assessed through ESM using a single question ("*Where are you right now?*") with 16 answer choices. The "*environment*" variable was then coded as "*familiar environment*" if the participant answered by referring to a personal environment (e.g., "*at home in my room,*" "*at home in a shared space,*" "*at work or in class*"). If the participant's response did not refer to a personal environment (e.g., "*in a restaurant or bar,*" "*in a store*"), the variable was coded as "*external environment.*"

**Table 1. Sample characteristics (n = 62).**

| Type of variables | Variable | Level | M or % | SD or n |
|---|---|---|---|---|
| Socio-demographics | Age (yrs) | | 15.19 | 1.53 |
| | Verbal IQ | | 11.03 | 2.98 |
| | Puberty | | 2.76 | 0.59 |
| | Socio-economic level | Low | 17.24 | 10 |
| | | Medium | 37.93 | 22 |
| | | High | 44.83 | 26 |
| | Nationality | Swiss | 82.25 | 51 |
| | First language | French | 90.23 | 56 |
| | Ongoing education | Yes | 95.16 | 59 |
| Within-person | mMean pos | | 76.22 | 22.72 |
| | mMean neg | | 15.28 | 18.00 |
| | smICC pos | | 14.15 | 5.65 |
| | smICC neg | | -1.39 | 3.09 |
| | Anger rumination | | 12.58 | 23.13 |
| | Self-control | | 74.17 | 21.81 |
| Between-person | gMean pos | | 76.78 | 16.59 |
| | gMean neg | | 15.35 | 10.58 |
| | gICC pos | | -0.86 | 0.45 |
| | gICC neg | | -0.46 | 0.32 |
| | Non adaptive ER | | 31.88 | 8.99 |
| | Self-control | | 40.84 | 6.99 |
| | Adjustment problems | | 56.45 | 7.14 |

*Note.* Verbal IQ: verbal intelligence quotient; *Between-person level:* gICC pos: global intra-class correlations for positive emotions (granularity); gICC neg: global intra-class correlations for positive emotions (granularity); gMean pos: Global mean score of positive emotions (intensity of positive emotions through the whole ESM measures); gMean neg: Global mean score of negative emotions (intensity of negative emotions through the whole ESM measures); SD pos: Standard deviation of positive emotions through the whole ESM measures (variability); SD neg: Standard deviation of negative emotions through the whole ESM measures (variability); Age (yrs): the participant's age in years; Verbal IQ: verbal intelligent quotient; ER: emotion regulation. *Within-person level*: mICC pos: momentary intra-class correlations for positive emotions (granularity); mICC neg: momentary intra-class correlations for positive emotions (granularity); smICC pos: root-squared transformed momentary intra-class correlations for positive emotions (granularity); smICC neg: root-squared transformed momentary intra-class correlations for negative emotions (granularity); mMean pos: Momentary mean score of positive emotions (intensity of positive emotions); mMean neg: Momentary mean score of negative emotions (intensity of negative emotions).

The "*presence of others*" variable referred to momentary social context and comprised three categories: "*alone,*" "*family/friend,*" "*others.*" It was assessed via ESM using a single question ("*Who is with you at the moment?*") with eight answer choices. The "*presence of other people*" variable was coded as "*alone*" if the participant answered "*nobody,*" as "*family/friend*" if the participant answered that they were in the presence of a friend or family member, and as "*other*" if they answered that they were in the presence of any other person.

**Emotion regulation—anger rumination.** Anger rumination (a specific non-adaptive emotion regulation strategy) was assessed via ESM using two questions ("*Since the last survey … I ruminated/thought about my past angry experiences*"; "*… I analyzed the events that made me angry*") rated on a visual analogue scale ranging from 0 ("*No, not at all*") to 100 ("*Yes, totally*"). These two questions were adapted from the anger rumination scale [46].

**Self-Control.** Self-control variability was assessed via ESM and measured by asking participants to rate four items ("*Since the last survey … I was able to focus on the activities at hand without being distracted*"; "*… I was able to follow my plans and objectives*"; "*… I lost control of myself*"; "*… I easily resisted the temptation/ my desires of the moment*") which were adapted from the revised early adolescent temperament questionnaire [47], the Barratt Impulsiveness Scale [48],

and the Brief Self-Control Scale [43]. Items were rated on a visual analogue scale ranging from 0 ("*No, not at all*") to 100 ("*Yes, totally*").

**2.4.3. Emotion dynamics features (at both between-person and within-person levels).** Emotions were measured by asking the participants to rate via ESM their experienced emotional intensity ("*Now, I feel…*") on a set of six emotion-related adjectives (three positive emotions—"*good*," "*quiet*," "*happy*"—and three negative ones—"*nervous/excited*," "*anxious/afraid*," "*angry/annoyed*"), using a visual analogue scale ranging from 0 ("*No, not at all*") to 100 ("*Yes, totally*"). Two main measures were computed from these items, namely, emotional intensity and differentiation.

**Emotional intensity.** Emotional intensity scores were obtained by calculating the positive and negative emotions in each survey (within-person level, momentary intensity) and across the entire sampling period (between-person level, global intensity). To do so, we computed means for positive (three items: *Now, I feel…* "*good*," "*quiet*," "*happy*") and negative emotions (three items: *Now, I feel…* "*nervous/excited*," "*anxious/afraid*," "*angry/annoyed*") separately, at either the within-person or the between-person level. Higher scores represent higher intensity.

**Emotion differentiation.** In line with previous studies [e.g., 49], differentiation at the between-person level (gICC) was estimated as an intraclass correlation (ICC) to assess the evaluations' agreement across the emotions at each measurement point [using the 'A-k' method; 50]. High ICCs indicate that emotional appraisals are relatively uniform, suggesting that across different measurement points, different emotional terms are used interchangeably to describe emotional experiences, reflecting low emotional differentiation. In contrast, low ICCs indicate variability in emotional appraisals, meaning that different emotional terms are used to describe emotional experiences, suggesting increased sensitivity to specific variations in emotional experiences and reflecting high emotional differentiation. At the within-person level, the momentary differentiation scores (mICC) were estimated using the momentary emotion differentiation index developed by Erbas et al. [30; available at https://osf.io/p25jv/], which is directly derived from the classical differentiation index (ICC). A high mICC (score approaching 0) suggests that the level of differentiation at that specific time point is high relative to the individual's overall level of differentiation (gICC).

## 2.5. Data analyses

All scores suited Gaussian-like distribution except the mICC scores. Thus, these scores were first inverted and then root-squared transformed (smICC) to reduce the skewness. At the end, we re-inverted them by multiplying them so that a higher smICC score reflects higher differentiation.

First, to assess the importance of the social environment (context and presence of others) to emotion dynamics features, we performed a 2 (social environment: familiar vs. external) x 3 (presence of others: alone, family/friend, other) multivariate analysis of variances (MANOVA) on the emotional intensity (mMEAN) and differentiation (smICC).

Then, to further examine the associations between the variables of interest (i.e., social environment, psychological characteristics, and emotion dynamics), we first computed Bravais–Pearson coefficients of correlation at the between-person and within-person levels. At the within-person level, we computed partial correlation controlling for the between-person-level variables. We then computed multilevel regression models including both within- and between-person variables for the associations of the social environment and psychological characteristics with emotional intensity (mMEAN) and differentiation (smICC). For instance, the formula for the momentary intensity of positive emotions (mMEAN pos) was computed as follows:

mMEAN pos =

First level (within-person): $\gamma_{10}$*Time + $\gamma_{20}$*Context (familiar vs. external) + $\gamma_{30}$*Presence of other (dummy1; alone: yes or no) + $\gamma_{30}$*Presence of others (dummy2; non-familiar other: yes or no) + $\gamma_{40}$* Momentary Anger rumination + $\gamma_{50}$*Momentary self-control + r0i+ et$_i$

Second level (between-person): $\beta_{00} + \beta_{01}*Age_i + \beta_{02}*Puberty_i + \beta_{03}*Non\ adapt\ emotion\ regulation_i + \beta_{04}*Trait\ self\text{-}control_i + \beta_{05}*Adjustment\ problems_i + \beta_{06}*Verbal\ IQ_i + r0_i + et_i.$

The multilevel models were computed separately for positive and negative emotions. The models were both based on the most appropriate mix between a priori hypotheses (top-down approach) and a posteriori results (bottom-up approach) of the correlational analyses. The analyses were computed with SPSS v27.0 and HLM 8.1.

According to the established rule [51], correlation coefficients of 0.10 are considered to have a small effect size; those of 0.30, a medium effect size; and those of 0.50 (and above), a large effect size. In the multilevel regression model, we used unstandardized beta (or gamma) coefficients, which indicate the change in the outcome variable for a one-unit change in the explaining factors, expressed in the original units of measurement. The main analyses referring to three models, the significance for the p-value was set at .017 (.05/3) to avoid Type I error.

## 3. Results

### 3.1 Role of social environment in emotion dynamics features

The results of the 2 (context: familiar vs. external) x 3 (presence of others: alone, family/friend, other) MANOVA revealed a significant main effect of context, $F(4, 1214) = 5.55$, $p < 0.001$, and a significant main effect of the presence of others, $F(4, 1214) = 4.05$, $p < 0.001$, but no interaction between both factors.

We observed a significant effect of context on the intensity of positive emotions, $F(1,1223) = 19.26$, $p < 0.001$, with more intense positive emotions in an external context (M = 79.06) compared to a familiar one (M = 71.82). Moreover, the analysis showed that negative emotions were more intense in a familiar context (M = 18.05) compared to an external one (M = 14.63), $F(1,1223) = 6.20$, $p = 0.013$ (see Fig 1, Panel A).

Although no effect of context was observed on the differentiation between positive emotions, we observed a significant effect of the context on the differentiation of negative emotions, $F(1,1223) = 6.63$, $p = 0.010$, with higher differentiation between negative emotions in an external context (M = -0.91) compared to a familiar one (M = -1.01) (see Fig 1, Panel B).

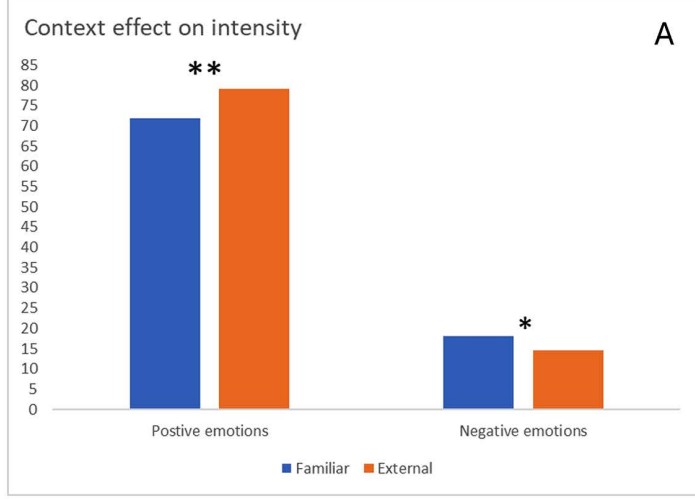
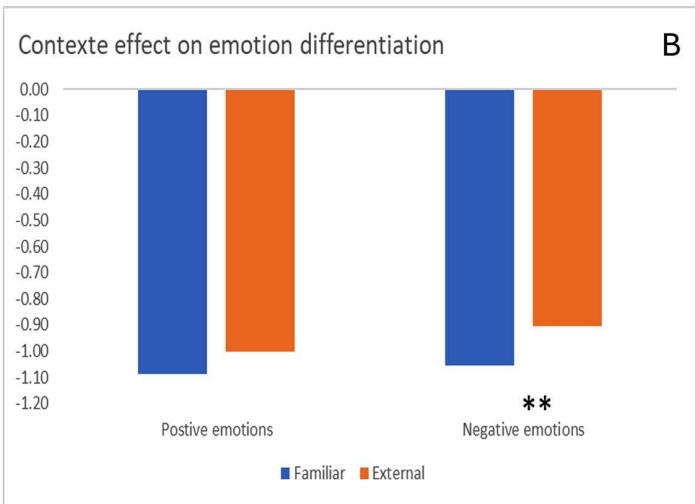

**Fig 1. Role of context on emotional intensity (Fig 1A) and differentiation (Fig 1B).** *Note.* *$p ≤ .05$; ** $p ≤ .01$; positive emotions = 'good', 'calm', 'happy'; negative emotions = 'nervous/excited', 'anxious/fearful', 'angry/annoyed'; Example of items for the familiar context: 'at home in my room', 'at home in my room'; Example of items for the external context: 'in a restaurant or bar', 'in a public building'.

The presence of others had a significant effect on the intensity of positive emotions, $F(2,1223)$ = 13.84, $p<0.001$. Post hoc tests with the least square differences (LSD) indicated that the intensity of positive emotions in the presence of a friend or family member (M = 80.74) was significantly ($ps<0.001$) higher than when the participant was alone (M = 75.72), or when another person was present (M = 69.87). The intensity of positive emotions was also significantly ($p=0.025$) higher when the adolescent was alone than in the presence of others.

The effect of the presence of others on the intensity of negative emotions was also significant ($F(2,1223)$ = 3.18, $p=0.042$). Post hoc tests with LSD correction indicated that the intensity of negative emotions in the presence of a friend or family member (M = 14.28) was significantly ($p=0.024$) lower than what was observed in the presence of another person (M = 18.78). On the other hand, there was no significant difference ($p≥0.145$) between the intensity of negative emotions in the presence of another person (a friend or family member, or another person) compared to when the participant was alone (M = 15.96) (see Fig 2).

No significant effect of the presence of others was observed on differentiation scores.

### 3.2. Correlation analyses

At the between-person level (Table 2), the intensity of positive and negative emotions was negatively correlated (large effect size). In addition, we observed that differentiation between positive emotions and differentiation between negative emotions were positively correlated (medium effect size) with each other. Higher differentiation of positive emotions was related to lower intensity of negative emotions (medium effect size). More adjustment problems were related to lower intensity of positive emotions and higher intensity of negative ones (both large effect sizes). The use of non-adaptive emotion regulation strategies was related to higher intensity of negative emotions (medium effect size), lower intensity of positive emotions (medium effect size), and lower differentiation of both positive (small effect size) and negative emotions (small effect size). A higher level of self-control was related to higher intensity of positive emotions (medium effect size)

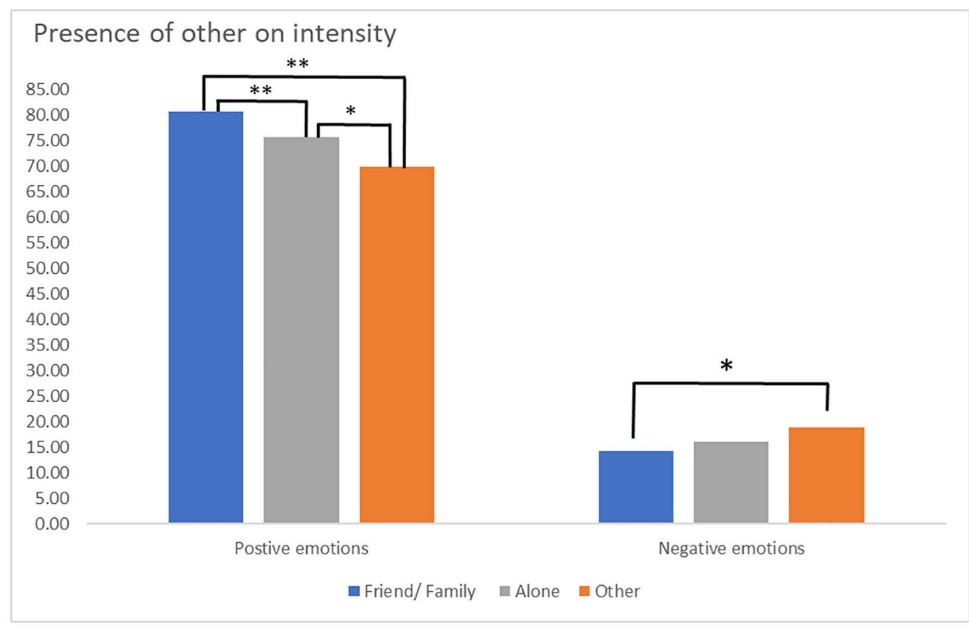

**Fig 2. Role of presence of others on emotional intensity.** *Note.* * $p<.05$, ** $p<.01$, corrected for least squares differences (LSD); Example of items for Family/ friend: 'family or partner', 'one or more friends'; Example of items for Alone: 'nobody'; Example of items for Other: 'Colleague or classmate', 'stranger', 'educator', 'carers', 'pet', 'other'.

**Table 2. Bravais-Pearson coefficients at the between-level person.**

|  | gMean pos | gMean neg | gICC pos | gICC neg |
|---|---|---|---|---|
| gMean neg | -.59* | – |  |  |
| gICC pos | .11 | -.31* | – |  |
| gICC neg | .06 | -.25 | 44** | – |
| Age | -.23 | .13 | -.17 | -.32* |
| SES | .05 | .01 | -.20 | -.12 |
| Verbal IQ | .02 | -.09** | .07** | .05* |
| Puberty | -.22** | .18** | -.20** | -.28** |
| Non adaptative ER | -.35** | .40** | -.11** | -.12** |
| Self-control | .48** | -.39** | .08** | .03 |
| Adjustment problems | -.57** | .52** | -.02 | -.06 |

*Note.* * p<.05, ** p<.01; gMean pos: Global mean score of positive emotions (intensity of positive emotions through the whole ESM measures); gMean neg: Global mean score of negative emotions (intensity of negative emotions through the whole ESM measures); gICC pos: global intra-class correlations for positive emotions (granularity); gICC neg: global intra-class correlations for positive emotions (granularity); Age (yrs): the participant's age in years; SES: socio-ecnomic status; Verbal IQ: verbal intelligence quotient; ER: Emotion regulation.

and less intensity of negative emotions (medium effect size) and to more differentiation between positive emotions (effect size below small cut off). Moreover, age was positively correlated with lower differentiation of negative emotions (medium effect size). SES was not related to any emotional features. Higher verbal IQ was associated with lower intensity of negative emotions (effect size below small cut off) and higher differentiation between both positive and negative emotions (effect size below small cut off). Puberty was positively associated with the intensity of negative emotions and negatively associated with the intensity of positive emotions and differentiation between both positive and negative emotions (small effect size).

At the within-person level (controlling for between-person level, Table 3), the momentary intensity of positive emotions was negatively related to the momentary intensity of negative emotions (medium effect size). In addition, differentiation between positive emotions and differentiation between negative emotions were positively correlated with each other (medium effect size). Time (across the 9 days) was related to lower momentary intensity of negative emotions, higher momentary differentiation of both positive and negative emotions, fewer momentary anger ruminations, and higher momentary self-control (all small effect sizes). Within a day (across the four assessments), adolescents had higher momentary intensity of positive emotions and lower momentary differentiation of positive emotions (both very small effect

**Table 3. Bravais-Pearson coefficients of correlation at the within-level person.**

|  | Time within the day | mMean pos | mMean neg | smICC pos | smICC neg | Anger rumination | Self-control |
|---|---|---|---|---|---|---|---|
| Days | -.01 | .02 | -.07* | .10*** | .09*** | -0.11*** | 0.05* |
| Time within the day | – | .07** | -.04 | -.06* | .02 | -.01 | -.01 |
| mMean pos |  | – | -.47*** | .34*** | .22*** | -.20*** | .38*** |
| mMean neg |  |  | – | -.30*** | -.49*** | .20*** | -.34*** |
| smICC pos |  |  |  | – | .34*** | -.041 | .15*** |
| smICC neg |  |  |  |  | – | -.033 | .14*** |
| Anger rumination |  |  |  |  |  | – | -.12*** |

*Note.* * p<.05, ** p<.01. *** p<.001. mMean pos: Momentary mean score of positive emotions (intensity of positive emotions); mMean neg: Momentary mean score of negative emotions (intensity of negative emotions); smICC pos: root-squared transformed momentary intra-class correlations for positive emotions (granularity); smICC neg: root-squared transformed momentary intra-class correlations for negative emotions (granularity).

sizes). Higher differentiation of positive and negative emotions was associated with higher momentary intensity of positive emotions and lower momentary intensity of negative emotions (medium to large effect size). Anger rumination was negatively correlated with momentary positive emotions and positively with the intensity of negative emotions (small effect size). Self-control was negatively correlated with the momentary intensity of negative emotions (medium effect size), whereas it was positively correlated with the momentary positive intensity of emotions (medium effect size) and differentiation of positive and negative emotions (small effect sizes).

### 3.3. Multilevel regression analyses

The multilevel models (Table 4) explained a significant part of the variance in intensity of positive emotions ($\chi^2(37) =$ 6,598.15, $p < 0.001$) and negative emotions ($\chi^2(37) = 171.57$, $p < 0.001$). The models computed on differentiation scores are not presented, since they were not reliable or significant and therefore not interpretable.

   At the between-person level, greater adjustment problems were related to lower positive emotions (decrease of 0.81 in positive emotions for each unit of increase in adjustment problems). Greater use of non-adaptive emotion regulation strategies was positively related to negative emotions (increase of 0.37 in negative emotions for each unit of increase in non-adaptive emotion regulation). A more advanced puberty stage was related to higher negative emotions (increase of 9.43 in negative emotions for each unit of increase in puberty), while older age was related to lower negative emotions (decrease of 3.79 in negative emotions for each unit of increase in age). At the within-person level, results revealed that being alone was associated with a lower intensity of positive emotions (decrease of 2.41 in positive emotions when being alone) and a higher intensity of negative emotions (increase of 2.40 in negative emotions when being alone). In the presence of an unfamiliar other, adolescents reported a lower intensity of positive emotions (decrease of 3.33 in positive emotions when being with unfamiliar others) and a higher intensity of negative emotions (increase of 4.59 in negative emotions when being with unfamiliar others). More anger rumination was related to lower intensity of positive emotions (decrease of 0.06 in positive emotions for each unit of increase in anger rumination). Higher states of self-control were related to higher intensity of positive emotions (increase of 0.28 in positive emotions for each unit of increase in self-control) and less intensity of negative emotions (decrease of 0.19 in negative emotions for each unit of increase in self-control).

**Table 4. Multilevel regression analyses on intensity.**

| Level | Outcomes: | mMean pos | | | mMean neg | | |
|---|---|---|---|---|---|---|---|
| | | β/ ϒ | S.E. | p-value | β/ ϒ | S.E. | p-value |
| Between-person | Age | -1.01 | 1.96 | .610 | **-3.79** | **1.10** | **.001** |
| | Puberty | 5.91 | 5.97 | .329 | **9.43** | **3.34** | **.008** |
| | Verbal IQ | -0.22 | 0.86 | .803 | -0.08 | 0.33 | .799 |
| | Non adaptive ER | -0.30 | 0.18 | .103 | **0.37** | **0.09** | **<.001** |
| | Trait self-control | -0.01 | 0.33 | .966 | -0.19 | 0.19 | .318 |
| | Adjustment Problems | **-0.81** | **0.39** | **.046** | -0.02 | 0.19 | .908 |
| Within-person | Time | 0.04 | 0.07 | .629 | -0.09 | 0.07 | .174 |
| | Context | 1.15 | 1.08 | .284 | -1.69 | 1.22 | .164 |
| | Alone | **-2.41** | **1.11** | **.030** | **2.40** | **1.14** | **.036** |
| | Non familiar others | **-3.33** | **1.70** | **.050** | **4.59** | **1.98** | **.021** |
| | Anger rumination | **-0.06** | **0.03** | **.034** | 0.05 | 0.04 | .273 |
| | State self-control | **0.28** | **0.04** | **<.001** | **-0.19** | **0.05** | **<.001** |

*Note.* mMean pos: Momentary mean score of positive emotions (intensity of positive emotions); mMean neg: Momentary mean score of negative emotions (intensity of negative emotions); ER: emotion regulation.

**Bold.** Significant results.

## 4. Discussion

This study examined the role of psychological characteristics and the social environment (i.e., context and presence of others) in the dynamics of six emotions in male adolescents' daily lives through a naturalistic approach. We observed that intensity and differentiation were related to each other, as observed in the literature [e.g., 52,53]. Specifically, our results revealed that non-adaptive emotion regulation strategies, such as rumination, dramatization, self-blaming, or blaming, are related to lower intensity of positive emotions, whereas higher self-control is associated with higher intensity of positive emotions and less intensity of negative emotions. Furthermore, we also identified the social environment's important role in emotional intensity and differentiation.

Adolescents who used non-adaptive emotion regulation strategies reported experiencing higher levels of negative emotions, replicating previous findings [e.g., 54]. However, at the within-person level, our results did not show that greater rumination was associated with greater intensity of negative emotions, although it was related to lower positive emotions. Thus, emotion regulation is more strongly associated with negative emotions at trait level, rather than at state level (at least specifically regarding anger rumination). In this vein, it should be noted that non-adaptive emotion regulation strategies are considered as transdiagnostic markers of psychopathologies [7].

Moreover, our results showed that, only at the state level (or within-person level), greater self-control was associated with a lower intensity of negative emotions, while it was associated with an increased intensity of positive emotions. In line with existing literature, we confirmed the important role of self-control in emotion dynamics features [24], but here we specify that this may occur more specifically in real time (at a state level) and not generally (at a trait level). Thus, the role of self-control in emotion dynamics occurs in real time, underlining the usefulness of self-control skills to adapt in the moment to the ever-changing environment [22].

In addition, greater adjustment problems were linked to less positive emotions. This result is consistent with the findings of a recent meta-analysis that identified lower overall positive (but not negative) emotion intensity as a key difference between youths with and without mental health problems [15].

Furthermore, with increasing age, adolescents showed lower levels of negative emotional intensity. This observation could support the hypothesis that, during the transition to young adulthood, the ability to self-regulate increases, which may be linked to the biological basis of maturation and the prefrontal cortex's more prominent role in subcortical circuitry [55–59]. Specifically, this may promote the modulation of emotional responses through emotion regulation and cognitive control [60]. However, it should be noted that several studies of wider age ranges have shown that negative emotional intensity remains stable during adolescence and even increases [15]. Consequently, these results must be interpreted with caution, as they may represent changes that occurred at a specific point in the development process.

Moreover, boys with a more mature pubertal stage had higher levels of negative emotions. This result may be understandable, as puberty is a stressful period [61]. In addition, alterations in brain regions involved in emotional processing, induced in part by neuroendocrine changes occurring during this period, suggest corresponding evolutions in emotional experience [62,63]. However, due to the limited number of studies examining puberty's influence on emotional intensity and the fact that these studies used different methodologies and presented contradictory results, it is difficult to formulate definitive conclusions [64]. It therefore seems essential to undertake fine-grained research to assess how emotional experiences evolve throughout the stages of puberty. This may help to better understand the interactions between puberty and emotional experiences during adolescence [64].

Moreover, our results highlighted that when adolescents are not in a familiar environment, they experience more positive emotions and fewer negative ones and are better able to differentiate between negative emotions. Complementarily, we observed that when adolescents are surrounded by friends or family members, they experience more positive emotions and less negative ones. These results are in line with previous studies, which show distinct trends depending on the social context [27,28,65]; however, they still may need additional exploration to better understand the role of the familiarity of the social environment and close persons. These observations can also support the idea

that positive emotions reflect active social engagement [66]. Indeed, the presence of friends or family members can create an environment of social support and affiliation, favoring an increase in the intensity of positive emotions [67] or helping to regulate and reduce the intensity of negative ones [68]. It is important to note that the presence of an unfamiliar person may not provide the same level of emotional support as the presence of a familiar person, which could explain the differences in negative emotional intensity observed between the presence of friends or family and the presence of a stranger.

Taking a broader perspective, we can speculate about how the Gross's process model of emotion regulation [e.g., 20,69,70], offers valuable insights on our results. Gross's model provides a framework for understanding emotion regulation across different stages: situation selection, situation modification, attentional deployment, cognitive change, and response modulation. In the situation selection phase, based on our data which showed that being alone was associated with lower positive emotion intensity and higher negative emotion intensity), we might hypothesize that adolescents choose familiar environments or social interactions (especially with family and friends) to enhance positive emotions. This may imply that adolescents might select social contexts as a strategy for emotion regulation. Regarding the situation modification phase, our findings suggest that external contexts are linked to a higher differentiation of negative emotions, suggesting that altering one's environment may be associated with emotional experiences. One hypothesis may be that adolescents might modify their surroundings to increase emotional granularity, either by seeking out new environments or by returning to familiar one to promote comfort. In the attentional deployment phase, our results on momentary self-control can be interpreted in the light of this framework. Indeed, higher momentary self-control was associated with greater positive emotion intensity and reduced negative emotion intensity. We can, thus, hypothesize that adolescents with better self-control may be more adept at directing their attention away from negative stimuli or toward positive aspects of their environment. At the cognitive change stage, our findings reveal that greater use of non-adaptive strategies (at the trait level) correlates with increased negative emotions, suggesting that adolescents who struggle with cognitive reappraisal or other adaptive strategies may experience more negative affect. This suggests that developing effective cognitive change techniques in emotion regulation might be helpful. By interpreting our results in the light of the Gross's model of emotion regulation, we may gain some insights of how adolescents' emotional experiences and regulation strategies correspond to different stages of emotion dynamics.

Our findings, thus, may open new avenues in developing preventive interventions or personalized treatments such as ecological momentary interventions or just-in-time adapted interventions [for a scoping review see 71] to enhance adolescents' mental health. For instance, by identifying youths who spend time alone, we may encourage them to establish contact with friends or to proactively initiate contact with significant others. Moreover, given the important role played by self-control and emotion regulation strategies, these processes deserve to be targeted and enhanced through specific treatment [for a scoping review see 72]. Globally, these strategies may improve emotional experiences and, consequently, enhance youths' well-being and may reduce loneliness, which is closely related to mental health [73–75].

This study suffers from several limitations that should be mentioned and that limit the generalizability of the results. Our sample, rather small, was solely composed of male adolescents with a relatively wide age range. Further studies may be conducted to generalize our results to more diverse populations in terms of gender and study more specifically the role of age. Furthermore, we assessed a restricted range of emotions, limiting ourselves to six emotions. Other emotions not included in our assessment could play an important role in emotion dynamics. Another important limitation to consider is that our protocol was only implemented on weekdays. It is therefore essential to recognize that emotional experiences on weekends may differ from those observed during the week. Consequently, our results may not exhaustively reflect the full range of emotional experiences in everyday life. Finally, as this study examined simultaneous associations, we are limited to conclusions about the way emotion intensity and differentiation are related to emotion regulation strategies, and we cannot support causality.

## 5. Conclusion

In conclusion, we observed associations at the trait level between adjustment problems, emotion regulation, and emotional experience, while self-control appeared to play a more specific role in real-time (i.e., state level) emotional dynamics. Finally, we observed specific developmental trends. In addition, our results highlight the crucial importance of the social environment and, in particular, the presence of others in the construction of the male adolescent identity, which underlies the dynamics of emotion. Therefore, our study provides important insights for a better understanding of the dynamics of emotion in the daily life of male adolescents.

## Supporting information

**S1 Table. Estimated marginal means of positive and negative emotions across personal and external environments.**
(DOCX)

**S2 Table. Estimated marginal means of positive emotion expression across presence types.**
(DOCX)

**S1 Fig. Role of context on emotional intensity.**
(DOCX)

**S2 Fig. Role of context on emotional granularity.**
(DOCX)

**S3 Fig. Role of presence of others on intensity of positive and negative emotions.**
(DOCX)

## Acknowledgments

We would like to acknowledge the participants who took part in the study. Moreover, we are thankful to their parents or the residential care institution that helped us recruit the youths.

## Author contributions

**Conceptualization:** Kerstin Jessica Plessen, Sebastien Urben.

**Formal analysis:** Umberto Cauzo, Jennifer Glaus, Sebastien Urben.

**Funding acquisition:** Sebastien Urben.

**Investigation:** Lauriane Constanty, Giorgia Miano, Sebastien Urben.

**Methodology:** Kerstin Jessica Plessen, Sebastien Urben.

**Project administration:** Sebastien Urben.

**Resources:** Kerstin Jessica Plessen, Sebastien Urben.

**Supervision:** Kerstin Jessica Plessen, Sebastien Urben.

**Validation:** Umberto Cauzo, Lauriane Constanty.

**Writing – original draft:** Umberto Cauzo, Sebastien Urben.

**Writing – review & editing:** Umberto Cauzo, Lauriane Constanty, Jennifer Glaus, Julia Giovannini, Marion Abi Kheir, Giorgia Miano, Caroline Lepage, Kerstin Jessica Plessen, Sebastien Urben.

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
