## [Decision Letter · Decision Letter 0]

4 Jun 2025

PMEN-D-25-00088

Examining Emotion Dynamics in Daily Life in Male Adolescents: An Experience Sampling Method Study

PLOS Mental Health

Dear Dr. Urben,

Thank you for submitting your manuscript to PLOS Mental Health. After careful consideration, we feel that it has merit but does not fully meet PLOS Mental Health’s publication criteria as it currently stands. Therefore, we invite you to submit a revised version of the manuscript that addresses the points raised during the review process.

EDITOR: Be sure to:

Critically address the changes raised by the reviewers

Please submit your revised manuscript by **3rd July 2025.** If you will need more time than this to complete your revisions, please reply to this message or contact the journal office at mentalhealth@plos.org. Please include the following items when submitting your revised manuscript:

We look forward to receiving your revised manuscript.

Kind regards,

Kizito Omona, PhD

Academic Editor

PLOS Mental Health

Journal Requirements:

1. Please provide additional details regarding participant consent. In the ethics statement in the Methods and online submission information, please ensure that you have specified (1) whether consent was informed and (2) what type you obtained (for instance, written or verbal, and if verbal, how it was documented and witnessed). If your study included minors, state whether you obtained consent from parents or guardians. If the need for consent was waived by the ethics committee, please include this information.

2. We ask that a manuscript source file is provided at Revision. Please upload your manuscript file as a .doc, .docx, .rtf or .tex.

3. We do not publish any copyright or trademark symbols that usually accompany proprietary names, eg (R), (C), or TM (e.g. next to drug or reagent names). Please remove all instances of trademark/copyright symbols throughout the text, including ® on page 5.

4. We note that your Data Availability Statement is currently as follows: [Data is provided within the manuscript. Further inquiries can be directed to the corresponding authors.]

Additional Editor Comments (if provided):

Critically address the changes raised by the reviewers

Reviewers' comments:

Reviewer's Responses to Questions

**Comments to the Author**

1. Does this manuscript meet PLOS Mental Health’s publication criteria?

Reviewer #1: Yes

Reviewer #2: Yes

2. Has the statistical analysis been performed appropriately and rigorously?

Reviewer #1: Yes

Reviewer #2: Yes

3. Have the authors made all data underlying the findings in their manuscript fully available (please refer to the Data Availability Statement at the start of the manuscript PDF file)?

Reviewer #1: Yes

Reviewer #2: Yes

4. Is the manuscript presented in an intelligible fashion and written in standard English?

Reviewer #1: Yes

Reviewer #2: Yes

Reviewer #1: Major Points:

This study uses an experience sampling method to examine emotion dynamics in male adolescents, but there are several methodological and interpretive concerns that should be addressed before publication.

The correlation coefficients reported in Tables 2a and 2b include values as small as -0.02 that are marked as statistically significant. While these may clear the p-value threshold due to the large sample size (62 participants × 9 days × 4 measurements = ~2,200+ observations), the practical significance of such tiny correlations is questionable. A correlation of -0.02 indicates essentially no relationship between variables. I strongly recommend that you distinguish between statistical and practical significance throughout the manuscript and consider using effect size interpretations (e.g., Cohen's guidelines where r=.10 is small, r=.30 is medium, and r=.50 is large).

The abstract mentions "higher differentiation between negative emotions" in an external context, but the multilevel models computed on differentiation scores were "not reliable or significant and therefore not interpretable" (p. 13). This inconsistency needs to be resolved - you can't claim a finding in the abstract that you later state wasn't reliable.

There appears to be a disconnect between your research question and your conclusions. You initially aimed to examine "how psychological characteristics and the social environment influence the dynamics of emotions" but many conclusions stretch beyond what the data support, especially regarding causal inferences.

Additional Issues:

Sample limitations need more acknowledgment. The exclusion of female participants severely limits generalizability, especially since you note gender differences in emotion dynamics. The age range of 12-17 is also quite broad considering the rapid developmental changes in this period.

The abstract is missing crucial methodological details. It states "62 male adolescents with varying degrees of adjustment problems," but doesn't clarify how these problems were measured or the inclusion criteria.

Your discussion on p.17-18 about Gross's process model of emotion regulation is speculative and goes beyond your data. You state "our findings align with Gross's process model" but then make numerous interpretive leaps without direct evidence.

The conclusion mentions "a stable influence of adjustment problems and emotion regulation" but your design can't determine stability since it's not longitudinal in the traditional sense.

Your title suggests a comprehensive examination of emotion dynamics, but you only measured six emotions (three positive, three negative), which is a significant limitation that should be acknowledged more prominently.

The multilevel regression analyses (Table 3) show several significant coefficients that are very small in magnitude. For example, non-adaptive emotion regulation shows a significant coefficient of 0.37 for negative emotions - what does this mean in real-world terms? The practical significance of these findings is unclear.

The theoretical framing focuses heavily on maturation of the prefrontal cortex versus limbic system, but the study includes no neuroimaging measures. This creates a disconnect between your theoretical framework and methodology.

There are several places where you make causal claims (e.g., "higher states of self-control were related to higher intensity of positive emotions and less intensity of negative ones") when your design only supports correlational relationships.

The limitations section on p.17-18 doesn't address the issue of multiple statistical tests without proper correction, which increases the risk of Type I errors.

You state that data is "provided within the manuscript" but it's unclear if the raw data will be accessible to other researchers, which is increasingly an expectation for publication.

Summary:

This manuscript examines an important topic but has significant issues with the interpretation of results, particularly regarding small correlation coefficients that achieve statistical significance but lack practical importance. The abstract and conclusions make claims that aren't fully supported by the data, and there's insufficient acknowledgment of the limitations of the study design for making causal inferences.

Reviewer #2: Title: Examining Emotion Dynamics in Daily Life in Male Adolescents: An Experience Sampling Method Study

This manuscript addresses a relevant and timely topic in adolescent psychology using an experience sampling method (ESM), which is suitable for capturing emotion fluctuations in real-life settings. The study focuses specifically on male adolescents, offering potential insights into gender-specific emotional processes. While the paper contributes to the growing field of emotion dynamics, there are areas that limit its overall impact.

Strengths:

- The use of ESM enhances ecological validity.

- The research question is clearly stated and addresses a gap in current literature.

- The manuscript is generally well-structured and readable.

Limitations:

- The sample size is relatively modest, which may limit generalizability.

- Some statistical analyses and interpretations could benefit from further clarification or justification.

- The discussion section could better connect findings to broader theoretical frameworks

**Do you want your identity to be public for this peer review?** For information about this choice, including consent withdrawal, please see our Privacy Policy

Reviewer #1: **Yes: ** Ahmet Erhan Bakırcı

Reviewer #2: **Yes: ** Tala jihad Aldarabkeh

---

## [Decision Letter · Decision Letter 1]

22 Sep 2025

PMEN-D-25-00088R1

Examining Specific Emotion Dynamics in Daily Life in Male Adolescents: An Experience Sampling Method Study

PLOS Mental Health

**Dear Dr. Urben,**

Thank you for submitting your manuscript to PLOS Mental Health. After careful consideration, we feel that it has merit but does not fully meet PLOS Mental Health’s publication criteria as it currently stands. Therefore, we invite you to submit a revised version of the manuscript that addresses the points raised during the review process.

EDITOR: Be sure to:

**Consent:** You stated that “each adolescent and their legal representative completed a written consent form”, but you did not clarify how comprehension was ensured and whether assent was obtained separately from minors. Revise and clearly state these; comprehension and assent for minors**Data Sharing:** Your current data availability statement does not comply with the PLOS Data Policy. It is not acceptable for the corresponding author to be the sole gatekeeper of the data. If some participants did not consent to data reuse, the minimal dataset from participants who did consent should still be shared. Publish your dataset in a public repository after removing the identifiers. You can use Harvard Dataverse or Mendeley or any other and show the digital object indentifier (DOI) of the dataset. 

Please ensure that your decision is justified on PLOS Mental Health’s publication criteria  and not, for example, on novelty or perceived impact.

We look forward to receiving your revised manuscript.

Kind regards,

Kizito Omona, PhD

Academic Editor

PLOS Mental Health

Journal Requirements:

Reviewers' comments:

Reviewer's Responses to Questions

**Comments to the Author**

Reviewer #3: All comments have been addressed

Reviewer #4: All comments have been addressed

publication criteria?

Reviewer #3: Yes

Reviewer #4: Partly

3. Has the statistical analysis been performed appropriately and rigorously?

Reviewer #3: I don't know

Reviewer #4: Yes

4. Have the authors made all data underlying the findings in their manuscript fully available (please refer to the Data Availability Statement at the start of the manuscript PDF file)?

Reviewer #3: Yes

Reviewer #4: No

5. Is the manuscript presented in an intelligible fashion and written in standard English?

Reviewer #3: Yes

Reviewer #4: Yes

Reviewer #3: An interesting study with an important aim to explore the effects of social environment and individual psychological characteristics on adolescent mental health and development. The vital age range of 12 to 17 years was chosen for this research study group. This is a wise choice because it is a stage of increased development of social intelligence, brain function, and reasoning, as well as psychological changes, which ultimately shape adult behaviors in male adolescents. The authors focused on 2 main features: emotional intensity and emotional granulation. They highlighted previous study reports on emotion regulation and further explained how these are categorized as adaptive and non-adaptive, with an emphasis on the positive and negative effects of utilizing either emotional category. They further explained how environment can contribute to fostering stronger positive emotions and adolescents e.g. types of social environment and the presence of others in the environment. Emphasis was placed on how there has been a lack of previous studies which addressed this vital state of adolescent development. Personally, as a clinician and study reviewer, I find it rather unfortunate because it is common knowledge that adolescents who learn to appropriately regulate their emotions by managing their stress and frustrations tend to develop better adult coping mechanisms. The authors employ the EMS (experienced sample method) for real-time emotional measurements in natural environments, with a focus on male subjects. The study was authorized by an Ethics Committee, and written consent was obtained from the study participants and their legal representatives. Study participants' exclusion was well detailed (diagnosis of mental health disorder and autism spectrum disorders). The various study measures and how these measures were obtained were well detailed. The authors presented well-structured statistical results with correlation analysis. This is a very important study that further substantiates the common knowledge that adolescent emotional experiences, characteristics, and environmental exposure contribute to the views that adolescent males have of themselves. Positive emotional States and a supportive environment tend to foster stronger self-esteem, while non-adaptive emotions, such as rumination, dramatization, and self-blaming, can negatively affect adult self-concept and resilience. This is an important study worthy of publication as an educational resource for identifying and further substantiating how the emotional states of adolescent males and social environment play a vital role in influencing personality development, relationship building, and coping skills in both young and older male adults.

Minor revision suggestion: the last paragraph on page 16 may need a minor adjustment. It states that

“Moreover, our results highlighted that when adolescents are NOT in a familiar environment, they experience more positive emotions and fewer negative ones and are better able to differentiate between negative emotions”

The word …. “NOT”, in the first sentence of that paragraph should be removed so that the study conclusion will better correlate with the study details.

Thank you so much.

Reviewer #4: The revised version of your manuscript shows clear improvement. You have appropriately adjusted the title, clarified some methodological issues, and moderated causal interpretations. The abstract is more consistent with the reported results, and the inclusion of effect size interpretations is welcome. These steps strengthen the manuscript.

However, some important concerns remain before the paper can be considered for publication. These mainly relate to ethical transparency and data availability.

1. You state that “each adolescent and their legal representative completed a written consent form.”. Detail how comprehension of the study was ensured and whether any incentives were provided.

2. The current statement does not fully comply with PLOS requirements. While you mention that raw data cannot be fully shared due to ethical restrictions, the policy requires either:

(a) Minimal Data Set or (b) a clear process through which a data access committee or institutional body can grant access under restrictions. Please note it is not acceptable for an author to be the sole named individual responsible for ensuring data access. Read PLOS Data policy.

Merely stating that the corresponding author can be contacted is insufficient. Please specify exactly which data are publicly available, what is restricted, why, and how researchers may formally request access.

**Do you want your identity to be public for this peer review?** For information about this choice, including consent withdrawal, please see our Privacy Policy

Reviewer #3: **Yes: ** Nonye Tochi Aghanya MSc, RN, FNP-C

Reviewer #4: No

---

## [Decision Letter · Decision Letter 2]

24 Nov 2025

Examining Specific Emotion Dynamics in Daily Life in Male Adolescents: An Experience Sampling Method Study

PMEN-D-25-00088R2

Dear Dr. Urben,

We are pleased to inform you that your manuscript 'Examining Specific Emotion Dynamics in Daily Life in Male Adolescents: An Experience Sampling Method Study' has been provisionally accepted for publication in PLOS Mental Health.

Best regards,

Kizito Omona, PhD

Academic Editor

PLOS Mental Health

Thank you for addressing the comments raised by the reviewers

Reviewer Comments (if any, and for reference):

Reviewer's Responses to Questions

**Comments to the Author**

Reviewer #3: (No Response)

Reviewer #5: All comments have been addressed

publication criteria?

Reviewer #3: Yes

Reviewer #5: Yes

3. Has the statistical analysis been performed appropriately and rigorously?

Reviewer #3: I don't know

Reviewer #5: Yes

4. Have the authors made all data underlying the findings in their manuscript fully available (please refer to the Data Availability Statement at the start of the manuscript PDF file)?

Reviewer #3: Yes

Reviewer #5: No

5. Is the manuscript presented in an intelligible fashion and written in standard English?

Reviewer #3: Yes

Reviewer #5: Yes

Reviewer #3: The study's aim is focused on 2 main features of emotional dynamics: emotion intensity and emotional differentiations. This is a very important study to help differentiate developmental emotional adjustments and behaviors in adolescent male. This can assist professionals to zone in and identify the most appropriate emotional regulation strategies for each corresponding emotion. The authors further detailed out the investigative study aim of identifying role of social environment and psychological characteristics in the daily lives of adolescents' emotional dynamics. The study method was clear: use of assessment measures with specified administration times and study duration.

Ethical approval and participation consents were obtained. The authors disclosed how they captured emotional intensity scores and the class correlations that signify emotion differentiations.

Study limitations were defined to support the hypothesis that adolescent boys exhibit increased self-control and regulation during the transition to young adulthood. On the contrary, the authors noted that this was not the finding in other studies with wider adolescent age ranges, different from the focused age range in the study.

Due to the influence of rapidly changing times, cultural implications, and the result of different emotional engagements by the most impressionable members of society (the adolescent male), this is a very important study to help streamline educational resources for identifying and addressing developmental trends of the adolescent male in an ever-shifting economic and political atmosphere.

Thank you for the invitation to review this study.

Reviewer #5: Thank you for an interesting manuscript and a thorough experiment.

**Do you want your identity to be public for this peer review?** For information about this choice, including consent withdrawal, please see our Privacy Policy

Reviewer #3: **Yes: ** Nonye Tochi Aghanya MSc, RN, FNP-C

Reviewer #5: No
